# The Effect of Strict State Measures on the Epidemiologic Curve of COVID-19 Infection in the Context of a Developing Country: A Simulation from Jordan

**DOI:** 10.3390/ijerph17186530

**Published:** 2020-09-08

**Authors:** Khalid A. Kheirallah, Belal Alsinglawi, Abdallah Alzoubi, Motasem N. Saidan, Omar Mubin, Mohammed S. Alorjani, Fawaz Mzayek

**Affiliations:** 1Department of Public Health, Medical School of Jordan University of Science and Technology, Irbid 22110, Jordan; kakheirallah@just.edu.jo; 2School of Computer, Data and Mathematical Sciences, Western Sydney University, Rydalmere 2116, NSW, Australia; O.Mubin@westernsydney.edu.au; 3Department of Pharmacology, Medical School of Jordan University of Science and Technology, Irbid 22110, Jordan; aaalzoubi28@just.edu.jo; 4Chemical Engineering Department, School of Engineering, The University of Jordan, Amman 11942, Jordan; m.saidan@ju.edu.jo; 5Department of Pathology and Microbiology, Medical School of Jordan University of Science and Technology, Irbid 22110, Jordan; Msalorjani@just.edu.jo; 6Division of Epidemiology, Biostatistics, and Environmental Health, School of Public Health, The University of Memphis, Memphis, TN 38152, USA; fmzayek@memphis.edu

**Keywords:** COVID-19, simulation, SEIR, SARS-CoV-2, Jordan, SIR, pandemic, epidemic

## Abstract

COVID-19 has posed an unprecedented global public health threat and caused a significant number of severe cases that necessitated long hospitalization and overwhelmed health services in the most affected countries. In response, governments initiated a series of non-pharmaceutical interventions (NPIs) that led to severe economic and social impacts. The effect of these intervention measures on the spread of the COVID-19 pandemic are not well investigated within developing country settings. This study simulated the trajectories of the COVID-19 pandemic curve in Jordan between February and May and assessed the effect of Jordan’s strict NPI measures on the spread of COVID-19. A modified susceptible, exposed, infected, and recovered (SEIR) epidemic model was utilized. The compartments in the proposed model categorized the Jordanian population into six deterministic compartments: suspected, exposed, infectious pre-symptomatic, infectious with mild symptoms, infectious with moderate to severe symptoms, and recovered. The GLEAMviz client simulator was used to run the simulation model. Epidemic curves were plotted for estimated COVID-19 cases in the simulation model, and compared against the reported cases. The simulation model estimated the highest number of total daily new COVID-19 cases, in the pre-symptomatic compartmental state, to be 65 cases, with an epidemic curve growing to its peak in 49 days and terminating in a duration of 83 days, and a total simulated cumulative case count of 1048 cases. The curve representing the number of actual reported cases in Jordan showed a good pattern compatibility to that in the mild and moderate to severe compartmental states. The reproduction number under the NPIs was reduced from 5.6 to less than one. NPIs in Jordan seem to be effective in controlling the COVID-19 epidemic and reducing the reproduction rate. Early strict intervention measures showed evidence of containing and suppressing the disease.

## 1. Introduction

Newly evolved coronaviruses (CoVs), such as the severe acute respiratory syndrome coronavirus (SARS-CoV) and the Middle East respiratory syndrome coronavirus (MERS-CoV), have posed global public health threats, including the 2003 outbreak in Guangdong, China, and the 2012 outbreak in the Middle East, respectively [1]. Similarly, SARS-CoV-2, an enveloped positive-sense RNA virus that infects humans [2], was initially reported as a localized pneumonia epidemic around December 2019, in China, before being declared as a pandemic by the World Health Organization (WHO) early 2020 [3,4]. COVID-19, the disease caused by SARS-CoV-2, is today a pandemic and of high priority. As of 21 June 2020, more than 8.5 million confirmed COVID-19 cases and about 500 thousand deaths have been recorded worldwide [5].

While COVID-19 can cause severe illness and death, many uncertainties exist. The full extent of the pandemic, especially in developing countries, the full clinical spectrum of illness, including the prevalence of mildly symptomatic cases [4], and the true case fatality rates [6], are not truly known. With 81% of infected cases developing only mild symptoms of COVID-19, it was suggested that many infected individuals with mild symptoms may not seek testing [7]. This adds to the uncertainty of COVID-19 [8], especially in developing countries with limited testing and treating capabilities, and may make the true case count as much as 10 times higher than reported [9]. Projecting case count, therefore, is an essential tool for public health response measures and health system management.

Globally, two vital non-pharmaceutical intervention (NPI) strategies have been identified to control the spread of an epidemic: mitigation and suppression. The former focuses on slowing the spread of the disease, but not necessarily stopping it, by reducing the healthcare demand peak and by protecting at-risk groups. Suppression, on the other hand, focuses on reversing the epidemic’s growth, reducing case numbers to low levels and maintaining that situation indefinitely [10,11].

In developed countries, these measures have been effective in controlling the spread of COVID-19 [10,12,13]. Such an effect has been assessed using mathematical modeling that simulated the spread of SARS-CoV-2 infection across the population and shaped control measures that might mitigate future transmissions [10,12,13,14,15,16,17,18,19,20,21]. One outcome of such a simulation is the predicted epidemic curve, representing the number of infections caused by the virus over time. Using a set of parameters, such simulation measures the impact of different interventions that can directly affect the predicted epidemic curve [21]. Mathematical modelling, therefore, presented itself as a powerful tool for understanding the transmission of COVID-19 and exploring different scenarios. However, using such modeling in developing countries, where healthcare systems are relatively weak, protective equipment is scarce, and poor testing and treatment capacities exist, is controversial [7,22,23].

The Hashemite Kingdom of Jordan, a country in the Middle East region, activated its initial national response to COVID-19 on February 27 by banning non-Jordanian travelers from high-risk countries from entering Jordan. On March 2, the first COVID-19 case was reported for a national arriving from Italy. In the same week, Jordan initiated a quarantine for arrivals from selected European countries. On March 15, a total of 12 new cases were reported. In response, more restrictions were imposed, where all educational institutions, tourism sites, cafes, and restaurants were ordered to close. All arriving passengers were then handled as suspected cases and immediately quarantined. By March 18, Jordan prohibited travel between governorates, suspended all flights, closed borders, suspended public transportation, closed commercial complexes, suspended non-emergency medical services, closed public and private sectors, implemented stay-at-home policy, and prohibited public, social, and religious events. Jordan then declared a national lockdown, a state of emergency, and imposed a curfew and mandated wearing face masks in public places, including cars. During the early couple of days of the curfew, a complete nationwide lockdown banned people from leaving their households. Citizens were then allowed five specific days to locally move around and walk, and neighborhood grocery stores were allowed to open between 10 AM and 6 PM. Driving was not allowed and moving between administrative geographic boundaries was permitted under emergency circumstances.

The number of newly reported COVID-19 cases in Jordan fluctuated between three and 42 daily cases (mean number of daily reported cases was 15 cases). As of May 1, the number of reported cases was 459 cases, including eight deaths. Cases seem to have clustered among persons within the same family and a limited number of cases have been identified to be of unknown origin. Testing, taking place during the time at which this manuscript was prepared, has been conducted randomly, regardless of symptoms, within each of the 12 Jordanian governorates and a limited number of cases have been identified using this approach. In early May, the number of local cases reached zero for about 10 days.

At this stage, it is necessary to simulate the COVID-19 epidemic curve in Jordan, especially for those with mild symptoms, as this will give an indication of the actual national situation. Without proper simulation of cases by clinical manifestation, decisions to re-open businesses will be arbitrary and not data driven. From this perspective, the current research attempted to mathematically simulate the ongoing trajectory of the COVID-19 outbreak in Jordan and to model the effect of national interventions utilizing real-time scenarios. The simulation of the COVID-19 outbreak could be of added value for public health response planning and future expectations. The current research will also advance our knowledge about COVID-19 in developing countries and the effect of publicized responses implemented with widespread adherence and support in Jordan.

## 2. Materials and Methods

A modified susceptible, exposed, infected, and recovered (SEIR) epidemic model [24] to simulate the spread of COVID-19 in Jordan was utilized. The SEIR model simulates the spread of an infectious disease, assuming that no births or deaths occur and that no new individuals are introduced. As such, each individual is initially assigned to each of the following disease states (deterministic compartments): susceptible (S), exposed (E), infectious (I) or recovered (R). The deterministic compartments in the SEIR model are fairly sophisticated quantitative mathematical models, yet are easily run utilizing public data and known disease characteristics [24]. We have modified the standard SEIR model by adding compartmental states that reflect the compartmental population and research needs. Our modified model categorized the Jordanian population into six deterministic compartments: susceptible, exposed, infectious pre-symptomatic (representing the total number of infections in Jordan), infectious with mild symptoms (i.e., not needing hospitalization), infectious with moderate to severe symptoms (i.e., needing hospitalization), and recovered.

In designing the modified simulation model, we assumed that an exposed individual may become infectious, pre-symptomatic, and then may progress to recovered, or progress to become either a mild or moderate to severe symptomatic individual, both of whom may then progress to recovered. The following brief shows the compartmental states applied in our study:

**Susceptible**: all of the non-immune population in our study (the entire Jordanian population).

**Pre-symptomatic**: population producing or showing no COVID-19 symptoms yet, albeit infectious [11].

**Symptomatic (mild or moderate to severe)**: population showing COVID-19 symptoms.

**Recovered**: population recovered from COVID-19 infection.

The modified model predicts the number of simulated COVID-19 cases by each compartmental state in Jordan. It also has the potential to distinguish hidden (asymptomatic or mild, not seeking hospital care) from identified infected cases needing hospitalization (moderate to severe cases). Indeed, standard SEIR models are estimated by assuming that all infected people are reported. Such an assumption for the novel COVID-19 pandemic is largely unreasonable, as many infected people show no symptoms or mild symptoms and, as the testing procedure is not available in mass, many remain undetected [25]. The model also accounts for hospitalization of moderate to severe cases by adjusting the contact rate. It is assumed that such cases will be detected and quarantined within a healthcare setting as they will be seeking medical services. Hence, their contact rates will decrease tremendously.

### 2.1. Simulated Model and Modelling Software

We have utilized the GLEAMviz client desktop application version 7 simulator [26] that combines world data such as country population and human mobility. The GLEAMviz elaborates compartmental stochastic models [27] for disease transmission in a pandemic event. Our analysis assumes that the first case entered Jordan on February 1, and the initial simulation started as such. A population size of 10.2 million was built into the client simulator. Moreover, the model allows for the limitation of mobility within the population and the restriction of travel as built-in functions within the designed models. The simulator provides rates within each compartment which were converted into numbers based on population size.

### 2.2. Model Parameterization

To run the simulation model, we utilized a series of parameters, as indicated by the simulator (Table 1) [26].

Our model does not provide estimates for the proportion requiring intensive care units (ICU) within hospitals nor the estimated number of COVID-19-related deaths. Providing these estimates requires details of the clinical fraction of infected people, the likelihood of clinical cases being severely ill, as well as a detailed understanding of the capacity of the health services in Jordan.

Two basic models were run to simulate the estimated numbers of COVID-19 cases by clinical manifestation, assuming two separate scenarios: the NPI scenario (S1), which was implemented in Jordan, and the no action scenario (S2). The former considered NPI implementation dates (starting March 17 and ending May 15), while the latter assumed no NPIs took place (see Appendix A). For each compartmental state, the number of simulated daily new COVID-19 cases was plotted. Accordingly, the epidemic curves are presented along with the duration of the epidemic (in days) and the time to the peak (in days). Each S1 curve was also fitted against the reported daily number of cases.

## 3. Results

Figure 1 presents the number of daily new COVID-19 cases in the pre-symptomatic compartmental state, simulated under the S1 and S2 curves using the same scale. The S1 curve is demonstrated as a “baby” curve under the S2 curve that started after February 1 and ended before April 20. The simulation model, under S1, predicted that on March 20 the highest number of daily new cases in the pre-symptomatic compartmental state would be 65 cases, after which the number of simulated daily new cases started to decrease. By April 24, the predicted daily new cases had leveled out to zero. Considering that the simulation was set to start on February 1, and the NPIs commenced on March 17, it took the epidemic curve 49 days to grow to its peak and the total duration of the epidemic curve was predicted at 83 days. The cumulative number of cases was predicted at 1048. For the hypothetical scenario of no action (S2), the epidemic took a total of 147 days to reach its peak of 238,142 daily new cases by June 27, and the cumulative number of cases reached about 9.5 million around December 1.

The simulated daily new mild COVID-19 cases under S1 reached their peak on March 21 with 36 cases and a total duration of 49 days (Figure 2), after which the simulated daily new mild case count started to decrease and reached, on April 27, zero daily new cases (the total duration of the epidemic curve was 87 days).

As seen in Figure 3, the simulated daily new moderate to severe cases, under S1, reached a maximum number on March 24 with a total of 46 cases (a total of 53 days). The number then decreased to zero cases on April 27 (the total number of days for the epidemic was 87 days).

In Figure 4, we plotted the actual reported daily new cases in Jordan against the simulated cases in our model (S1). The curves representing the simulated number of daily new COVID-19 cases, in both the mild and moderate to severe compartmental states, had good pattern compatibility with those depicting the number of reported cases in Jordan, with a peak of new cases on March 24.

Under S1, the simulated cumulative recovery was 1044 cases by June 30. Out of the total cumulative cases, 695 cases were in the moderate to severe compartmental state, i.e., needing hospital care, while 795 were in the mild compartmental state, i.e., mostly hidden cases within the community. Moreover, based on the S1 model, the simulated reproduction number (R_0_) for COVID-19 after implementing NPIs in Jordan was estimated at 0.9.

Further comparisons between the S1 and S2 simulated models are presented in the Appendix A.

## 4. Discussion

With COVID-19 imposing global public health and socioeconomic uncertainties, governments are counting on their people to adapt to NPIs in an effort to reduce the impact of the epidemic. The combined efforts of both the government and the people are then necessary to bring the epidemic under control. How people react and respond to the implemented NPI measures are critical to the epidemiological presentation of the epidemic. In this context, the current study assessed the effect of NPIs implemented in Jordan on the COVID-19 outbreak, utilizing simulation techniques. The simulated epidemic curves for COVID-19 provided evidence that Jordan may have successfully implemented NPI measures that facilitated suppressing (containing) the spread of the epidemic by reducing the number of daily new reported cases and the total duration of the epidemic. The effects of the adopted NPIs in Jordan on the number of daily new cases and the duration of the epidemic are even more appreciated when compared to the catastrophic effects of the hypothetical scenario of no action (see Appendix A). Our results suggest that swift, intensive, and targeted lockdowns in Jordan may have caused new COVID-19 cases to plummet and the health system to be protected. Our research therefore suggests that a strong containment policy implemented early on can combat the spread of a COVID-19 epidemic.

A recent study, which utilized statistical modeling based on Google reports on social distancing, assessed lockdown efficiency for 13 countries. Jordan, Italy, and Indonesia were categorized as countries with very high-level lockdowns. When correlating lockdown procedures and the infection rates, to assess the impact of lockdown policies on R_0_, Jordan was reported as a country with high lockdown efficiency for the period between February 15 and April 11. However, Italy and Indonesia were categorized as medium lockdown efficiency countries for the same period. Similarly, Germany and Spain were reported as “not gaining any productive results out of the lockdown procedures” for the same period, yet their efficiency levels improved between May and July. India, on the other hand, was reported to have a very strict lockdown policy yet was categorized, initially, to have a low lockdown efficiency. Later on, India was categorized to have a medium efficiency (between May and July). Late lockdown procedures detected in Brazil and the United States were reported to have a major impact on large outbreaks and to inversely contribute to elevated infection rates [39]. These results are in line with our simulated model for Jordan and suggest that the country has presented a successful strategy that allowed for the “snuffing” out of the COVID-19 pandemic at an early stage. Such success may be attributed to early adaptation to a complete national lockdown, early isolation of all arrivals and travelers for two weeks, and effective contact tracing through the already established crises management center, which facilitated centralized decision making.

The Jordan Ministry of Health is currently conducting a national seropositivity (Immunoglobulins M (IgM) and Immunoglobulins G (IgG)) study to assess the effect of its measures in combating the spread of COVID-19. This comes as a continuation of the random PCR testing conducted earlier by the Ministry after the first wave that we are simulating. The positivity rate of the PCR test for about 700,000 randomly collected samples was less than 0.03%, while the positivity rate for SARS-CoV-2 antibodies is less than 1% for about 500,000 tests performed so far (Jordan’s population is estimated at 10.2 million). The results of the latter are still being updated but the positivity rate seems to be in line with the reported national numbers and the random PCR testing results. Both results seem to point to the effectiveness of state measures in combating COVID-19 and to support our findings as well.

Strict NPI measures implemented in Jordan, which lasted for more than six weeks, appear to have reduced COVID-19 transmission and likely reduced the reproduction number to less than one. A similar discussion was presented for the UK, for example [13], where, in the absence of control measures, the epidemic would quickly overwhelm the healthcare system. A combination of moderate interventions (school closures, shielding of older groups and self-isolation) was predicted to be unlikely to prevent an epidemic that would far exceed the available ICU capacity in the UK. More intensive lockdown-type measures, however, predicted an effective protection of the healthcare system from being overwhelmed. Importantly, the lockdown scenario for the UK effectively reduced R_0_ to near or below one [13].

Our results are significant not only for public health decision makers, but also for risk communication and lessons learned. In case a new wave of the epidemic hits, the notion to initiate strict measures is supported by this model’s outcomes and would strengthen public messages to enhance the proper implementation of strict measures. This data-driven approach is vital to ensure population commitment and to, perhaps, aid the ongoing efforts of other countries with similar resources and culture.

In infectious disease epidemiology, sensitivity analysis provides an insight into how the uncertainty of the model inputs affects the model output, and which input tends to lead to variation in the output. The GLEAMviz simulation software application does not provide compartment modeling in the form of accessible algorithms. Therefore, the inputs of the compartments are the only parameters that can be controlled by the end user. This limited our abilities to examine the algorithm of GLEAMviz and to conduct sensitivity analyses. In a future study, our simulation could be further improved by introducing epidemiological compartmental models in the form of computational algorithms to be evaluated with a suitable sensitivity analysis. However, the Susceptible, Infected, and Recovered (SIR) original model is a standardized one that has been in use for several years in epidemic investigation. Importantly, when we were optimizing the model parameter values to facilitate a proper agreement between the simulated and reported COVID-19 cases (as presented in Figure 4), we were improving the validity of the model.

A combination of NPIs, isolation and contact tracing has been reported to present a synergistic effect that increased the prospect of containment of COVID-19 [40]. Knowing that Jordan has implemented strict contact tracing and isolation of contacts limits our ability to clearly compare the actual reported numbers to those presented under S1. Until detailed information about cases identified via contact tracing and isolation are made available, the presented model (S1) is the only available method to meet the objective of the current study. Moreover, the numbers presented under S2 seemed to be high values, as the scenario assumed that no prevention and control measures were implemented. Their interpretation, therefore, should be limited to a comparison with S1 and should be seen as mostly hypothetical.

The simulation presented in the current study has limitations. It was designed to monitor the evolution of the COVID-19 epidemic spread in Jordan utilizing parameters presented about the disease from the experience within developed countries. However, at this stage of the epidemic, country-specific parameters are not available. Furthermore, the contact rates used in the current simulation were generalized for the whole population and did not consider variability within households or local communities. The assumption of a universal contact rate used in the proposed model was, however, adjusted for all cases with moderate to severe clinical manifestations. Considering that these cases are most likely to be detected within healthcare settings and be hospitalized, we reduced their contact rate to its minimum to overcome this limitation.

Furthermore, recognizing co-morbidities within the population structure of Jordan and incorporating them within the compartmental states is assuredly of added value in this simulation. However, the reports from Jordan did not specify co-morbidities and only stated the number of cases. This is a limitation to the proposed model and limits our abilities to assess and compare, as stated before. However, our aim was to evaluate state measures and compare simulated numbers to reported ones. Future research should consider this population structure of comorbidities and fine tune the results to reflect such factors within simulation models.

## 5. Conclusions

NPIs in Jordan seem to be effective in controlling the COVID-19 epidemic and reducing the reproduction rate. Early strict intervention measures showed evidence of containing and suppressing the disease. 

## Figures and Tables

**Figure 1 ijerph-17-06530-f001:**
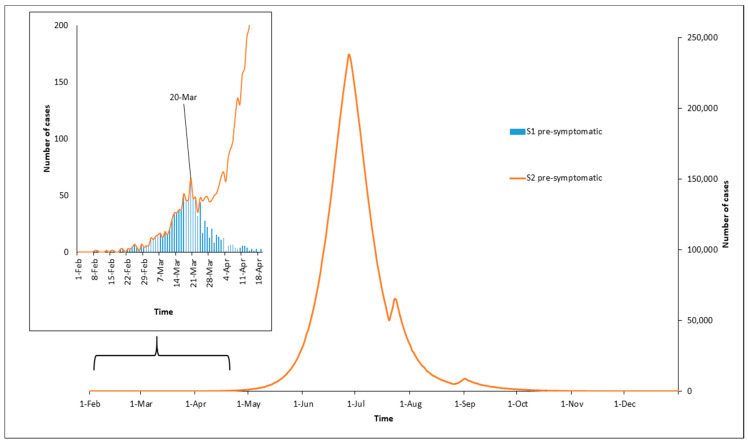
Simulated COVID-19 epidemic curves in Jordan under scenarios 1 and 2 (S1 and S2), utilizing the pre-symptomatic compartmental state.

**Figure 2 ijerph-17-06530-f002:**
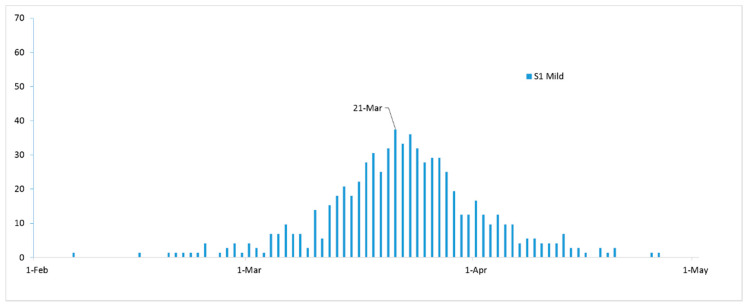
Simulated number of daily new COVID-19 cases in the mild compartmental state under scenario 1 (S1).

**Figure 3 ijerph-17-06530-f003:**
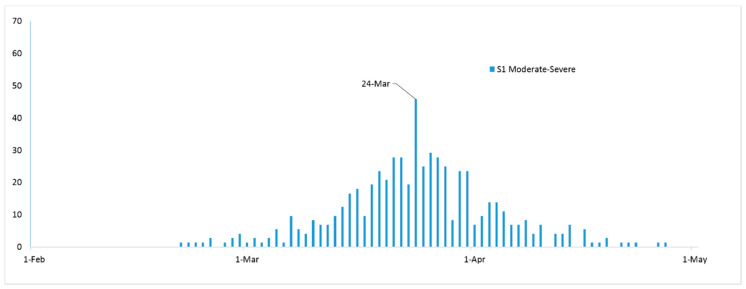
Simulated number of daily new COVID-19 cases in the moderate to severe compartmental state under scenario 1 (S1).

**Figure 4 ijerph-17-06530-f004:**
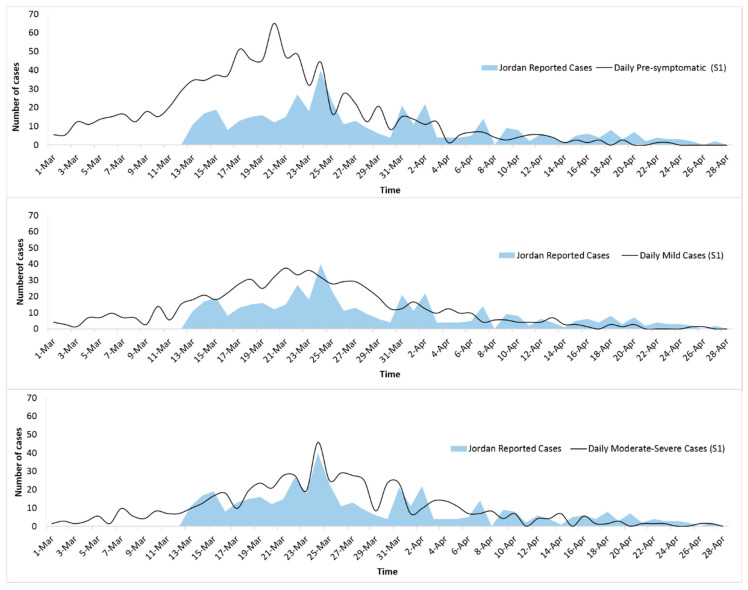
Number of daily new reported COVID-19 cases compared to S1-simulated numbers in the three compartmental states.

**Table 1 ijerph-17-06530-t001:** Model parameter descriptions and values used for simulating the number of COVID-19 cases in Jordan.

Parameter and Symbols	Description	Scenario 1 Values
β (beta)	Describes the transmission rate	February 1 to March 17 = 0.37 March 18 to April 24 = 0.06 April 25 to May 15 = 0.20 After May 15 = 0.37
α (alpha)	Reduction in transmission rate. (Moderate to Severe)	0.5
ε (epsilon)	The incubation period from the state of exposure to the disease to become infectious	1/5.2
*P_s_*	Probability of developing severe SAR-CoV-2 symptoms	0.01
*µ* (mu)	Recovery rate	1/14 days
*R_0_*	Basic Reproduction number	5.6

These parameters are as follows: Beta (β) describes the transmission rate and the spread of disease in the community. The β varies according to public health policies that are enforced or applied in communities such as pandemic containment, social distancing, remote working, closing schools, etc. Since Jordan’s culture is homogeneous, and people follow traditional forms for greeting, we have set the standard contact rate (β) to 0.37 [16,28,29]. To reflect the status of measures in Jordan, we added an extra layer (exception) to designate the non-pharmaceutical interventions (NPIs) that took place on March 17. As such, the contact rate value (β) was reduced from 0.37 to 0.06 [30] between March 17 and April 24. The contact rate value (β) was set to 0.2 between April 25 and May 15, reflecting the partial lifting of the curfew and partial reopening of selected businesses. After that, the contact rate value (β) was set to its original value of 0.37. Alpha (α) denotes the reduction in the transmission rate of hospitalized (moderate to severe) cases. We have used the value of α = 0.5 to reflect the negligible transmission rate of hospitalized patients. Epsilon (ε): the incubation period from the point of exposure to the disease becoming infectious. It is set to 5.2 days [9,10,31]. Ps: the probability of developing severe COVID-19 symptoms. This value was set at 0.01 [32]. Recovery rate (mu or *µ*), which indicates the time until an infectious case becomes recovered. Previous research [33] reports that the recovery time for COVID-19 is 14 days (*µ* = 1/14 days). Hence, we have used this value as the recovery rate (*µ* = 0.07) in our model. *R_0_*: the reproduction number for COVID-19. Based on the above values, *R_0_* was calculated as 5.6 (see Appendix A for formula). The basic reproduction number (*R_0_*) measures the transmission (contagious) potential of COVID-19 and describes the average number of secondary infections caused by a typical primary infection in a completely susceptible population. An *R_0_* value of 5.6 was reported in other similar global simulations [34]. The literature reported that *R_0_* ranges between 2.3 and 6.5 [28,35,36,37] and a re-analysis of Chinese data provided an updated estimate of 5.7 (95% CI 3.8–8.9) [37]. Other published studies reported that, for social gathering events such as wedding parties in Jordan, the *R_0_* value was five [38].

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
