# Peer review of "The Effect of Strict State Measures on the Epidemiologic Curve of COVID-19 Infection in the Context of a Developing Country: A Simulation from Jordan"

_ijerph, 2020, doi:10.3390/ijerph17186530_

Round 1
Reviewer 1 Report
(1) It needs some more information to understand Jordan's situation such as the number of tests compared to the number of populations, the proportion of positive among those who were tested, the proportion of patients whose transmission route was not confirmed, etc.
(2) Line 100: Does the random test performed mean that a certain number of the population was tested as a sample regardless of symptoms?
(3) Please reconfirm the description of each parameter in Table 1. The description of rate, period, number, etc. seems incorrect. Please explain the text in the order of the described parameters in the table.
(4) Ro was calculated relatively high. In Line 176, supplementary 1 for formula is described, but the supplementary table 1 shows the values applied to scenario 2 and supplementary figure 1 is a graph comparing S1 and S2. Please present the actual formula.
(5) Sensitivity analysis was not conducted?
Reviewer 2 Report
In addition to SEIR, recently a ball of other possible mathematical simulation models has been described. How the authors want to prove the superiority of the model they used. Or it can show that similar results will be obtained when we also apply a different simulation model
Reviewer 3 Report
This research is interesting however i consider that there is some information that should be available or that must there was a justification for its absence.
It is not clear, in the paragraph (line 80) when the measures were adopted in Jordan (line 86 onwards). Was also on March 15?
It is also not clear whether the use of masks was adopted (on lines 92 and 93 it is said that citizens were allowed to go shopping, but the specific conditions for doing so are not mentioned).
Shouldn't this kind of simulation be done taking into account the health status of the Jordanian population? It has been well studied that diabetes, hypertension, among others, can be risk factors and increase the number of symptomatic and hospitalized patients. Why were these variables not taken into account? Can't they change the models significantly?
In line 254, it is clear that Jordan has adopted a unique strategy! I don't think it was unique, there are other countries that have adopted very similar strategies.
The formatting of the entire document should be reviewed.
Finally, in paragraph 54, a global epidemic is a Pandemic. Thus, I am of the opinion that the word Pandemic should be used, as it is a reference in the WHO.
